# LENS: Localization enhanced by NeRF synthesis

**Arthur Moreau**[1,2]**, Nathan Piasco**[1]**, Dzmitry Tsishkou**[1]**, Bogdan Stanciulescu**[2]**, Arnaud de La Fortelle**[2]
[1]IoV team, Paris Research Center, Huawei [2]MINES ParisTech, PSL University, Center for robotics
arthur.moreau@mines-paristech.fr

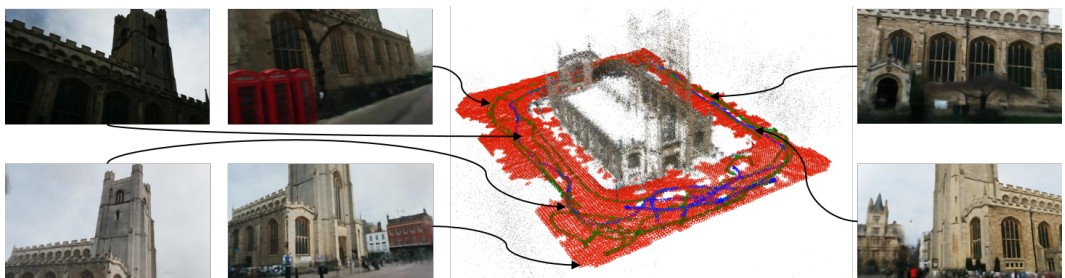

Figure 1: **Novel view synthesis with LENS:** given a 3D scene defined by a set of images labeled with corresponding poses (green cameras), we train a NeRF and render novel views (depicted images) on pose queries distributed across the scene (red cameras). Synthetic and real images are gathered to train a camera pose regression model, which performs twice better when evaluated on test set images (blue cameras) compared to the same model trained only on real training samples.

**Abstract:** Neural Radiance Fields [1] (NeRF) have recently demonstrated photo-realistic results for the task of novel view synthesis. In this paper, we propose to apply novel view synthesis to the robot relocalization problem: we demonstrate improvement of camera pose regression thanks to an additional synthetic dataset rendered by the NeRF class of algorithm. To avoid spawning novel views in irrelevant places we selected virtual camera locations from NeRF internal representation of the 3D geometry of the scene. We further improved localization accuracy of pose regressors using synthesized realistic and geometry consistent images as data augmentation during training. At the time of publication, our approach improved state of the art with a 60% lower error on Cambridge Landmarks and 7-scenes datasets. Hence, the resulting accuracy becomes comparable to structure-based methods, without any architecture modification or domain adaptation constraints. Since our method allows almost infinite generation of training data, we investigated limitations of camera pose regression depending on size and distribution of data used for training on public benchmarks. We concluded that pose regression accuracy is mostly bounded by relatively small and biased datasets rather than capacity of the pose regression model to solve the localization task.

**Keywords:** visual localization, camera pose regression, novel view synthesis

## 1 Introduction

In order to localize a robot or a vehicle in a known environment, one solution is to use camera images with visual based localization algorithms [2, 3]. These methods extract information from the image to predict camera position and orientation. Best performing methods in this field are structure-based methods: they solve the problem with pipelines that include feature extraction, image retrieval and/or 2D-3D feature matching [4, 5, 6]. However, their high computational cost and memory footprint make them difficult to deploy in real-time embedded systems for robotic and autonomous applications. In this setup, an alternative solution is to train a deep neural network to regress poses from images in an end-to-end supervised way [7]. These methods, named camera pose regressors,

5th Conference on Robot Learning (CoRL 2021), London, UK.

require an offline training step while being more efficient at test time [8]. However, despite many architectural improvements [9, 10, 8, 11], the accuracy provided by these networks is limited and highly dependent on quantity and diversity of training data [12].

Our hypothesis is that camera pose regression formulated as a machine learning problem is currently limited because training datasets are highly biased and lead to a poor generalization. In contrast with many tasks solved with deep learning, camera pose regressors are overfitted to a single scene and we expect their output space to be the set of possible camera poses in this scene. But in practice, training datasets for camera pose regression are often built from a limited number of consecutive video frames. Consequently, they lack diversity compared to the set of well distributed camera positions and orientations in a given scene (see figure 1). Sattler et al. [12] have shown that these networks are not able to extrapolate to unseen camera positions, and then struggle to perform better than an image retrieval baseline. We suppose that a training dataset which is well distributed on the entire scene should help to overcome this limitation.

Our proposal is to augment training datasets for pose regression using Neural Radiance Fields (NeRF) [1]. These networks perform novel view synthesis thanks to color and density predictions along camera rays and volume rendering techniques. Compared to standard generative models [13] used for view synthesis, NeRF rendering is tailored for our localization problem as it produces geometry-consistent images for any pose query in the scene thanks to the ray tracing approach. To optimize the pose regressor localization performances and mitigate the cost of images rendering, we propose an algorithm, named LENS, which generates virtual camera locations distributed on a regular grid across the scene. In order to avoid usage of degenerate views due to occlusions or wrong orientations, LENS leverages the internal scene geometry learned by the NeRF model. It allows to discard poses close to occluders like buildings, and provides only meaningful images to the pose regressor without any scene-specific parameter tuning. We apply our method on both indoor and outdoor datasets to show the benefit of a spatially balanced dataset for training a pose regressor.

In the next section, we review previous work related to novel view synthesis for visual localization. In section 3 we describe our method, specifically the choice of NeRF and the synthetic poses generation algorithm. Section 4 is dedicated to experiments conducted on Cambridge Landmarks and 7 scenes datasets. Section 5 concludes the paper.

## 2   Related work

Novel view synthesis can be used at several steps of a visual localization method. Zhang et al. [13] refine reference poses iteratively by comparing rendered view of the pose estimate with the original image. This idea can also be exploited in the relocalization step of a structure-based method: InLoc [14] verifies the predicted pose by comparing local patch descriptors between original and rendered image. iNeRF [15] performs gradient-based optimization to recover a pose estimate thanks to NeRF differentiability. Direct-PoseNet [16] adapts this idea to camera pose regression training by using an additional photometric loss between query image and NeRF synthesis on the predicted pose. PoseGan [17] learns jointly pose regression and view synthesis resulting in an improved localization accuracy.

Another direction is to use synthetic images to enlarge the reference database with more densely sampled views. Localization algorithms perform better when an image with a similar viewpoint is available for matching (structure based methods) or training (learning based). This idea can also be applied to other vision tasks, such as 3D semantic segmentation [18]. These methods yield two important design choices: how to chose where virtual cameras are located, and how to perform novel view synthesis.

Virtual camera locations are usually sampled using ad-hoc methods: Torii et al. [19] expand a place recognition database by sampling on a regular grid, removing locations that lie in buildings thanks to coarse depth plans representing basic 3D structures or locations that are too far from the original trajectory. Aubry et al. [20] and Irschara et al. [21] use a similar approach and discard cameras that doesn't view any portion of the 3d model. Some alternatives propose to use a probabilistic sampling strategy based on detectability of keypoints at a specific pose [22], or try to find an optimal set of locations with a genetic algorithm [23]. However, ad-hoc methods have been observed to be sufficient in practice [21], so our method uses a regular grid combined with NeRF internal representation of the scene to ensure meaningful views.

One popular approach to render novel view is to use a 3D textured mesh to represent the scene and to create virtual cameras within this scene [20, 12]. However, obtaining such scene representation is costly and not easy to compute from crowd-sourced images acquired in a dynamic environment. Meshs built from crowd sourced images usually fail to describe low textured or crowded part of the scene, such as sky and ground. Synthetic images can also be rendered thanks to Generative Adversarial Networks [22, 24]. Finally, recent methods that learn a continuous volumetric representation of the scene such as Neural Radiance Fields [1] outperform prior work and exhibit photorealistic results. To the best of our knowledge, no study has been conducted to enlarge camera pose regression training dataset with NeRF rendered views. We refer readers to the work of Shavit and Ferens [25] for further details about camera pose regression.

## 3 Synthetic dataset rendering with LENS

We provide here a description of our method, named LENS, which can be seen as an offline data augmentation pipeline to train pose regressors. Our goal is to generate a large and uniformly distributed dataset of synthetic images, using a small set of images depicting the scene labeled with reference camera poses (usually provided by structure from motion methods such as COLMAP [26]). This dataset is then used to train a camera pose regressor, resulting in an improved accuracy without additional computation during localization inference. Our framework is described in figure 2 and is detailed below.

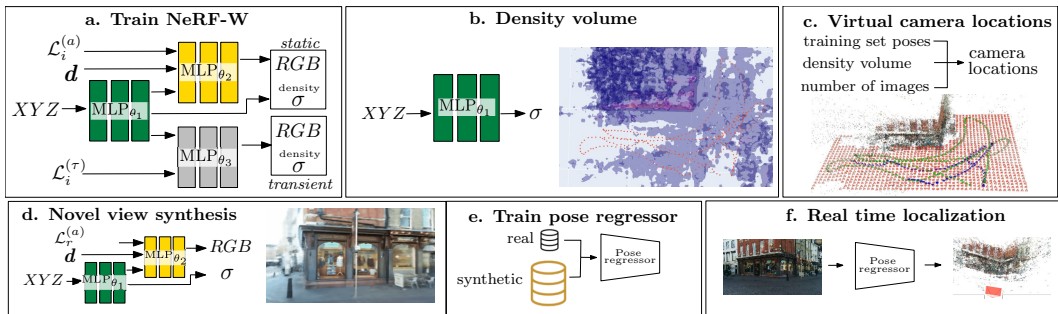

Figure 2: **LENS pipeline:** First, NeRF-W is trained with available real images (a.). Then, the trained model is used to detect high density points in the scene (b.). Poses from real images and high density locations are used to generate virtual camera locations (c.) on which novel view synthesis is performed (d.). Combined real and synthetic datasets are used to train a pose regressor (e.).

### 3.1 NeRF-W training

A neural radiance field learns a continuous 3D representation of a scene from a collections of images depicting this scene from known camera viewpoints. It uses 2 feed-forward neural networks: $MLP_{\theta_1}$ connects a 3D position to a density value $\sigma$ and $MLP_{\theta_2}$ predicts a RGB color from a viewing direction $d$ and a latent vector $MLP_{\theta_1}(x, y, z)$. The final RGB value of a pixel is computed by approximating the volume rendering integral using $N_c$ coarse samples and $N_f$ fine samples along a light ray with predicted colors $(c_i)_n$ and densities $(\sigma_i)_n$.

With this formulation, NeRF models are able to render a static scene captured under controlled settings but fail in real world dynamic scenes with variable illuminations and dynamic objects. NeRF in the Wild [27] (NeRF-W), overcomes this limitation by modelling these temporal modifications with 2 learned latent spaces $\mathcal{L}_i^{(a)}$ (appearance embedding) and $\mathcal{L}_i^{(\tau)}$ (transient embedding) that provide control on the appearance and dynamic content of each rendered view. An additional network $MLP_{\theta_3}$ is introduced to render scenes with transient objects during training thanks to $\mathcal{L}_i^{(\tau)}$, whereas $MLP_{\theta_2}$ only render the static part of the scene with $\mathcal{L}_i^{(a)}$ as an additional input, see figure 2-a. We train a NeRF-W model on each scene using a sparse set of registered images.

## 3.2 Density volume generation

In order to chose valid locations for the synthetic images, we first gather the volumetric representation of the scene learned by the NeRF-W model.

We query $\text{MLP}_{\theta_1}$ on a regular 3D grid that can be displayed as a density volume (see figure 2-b and figure 3). Lets consider $\mathcal{B}$ the smallest 3D bounding box that contains all the poses of the training images. The 3D grid extend is defined by another 3D bounding box $\mathcal{B}_+$ obtained by extending $\mathcal{B}$ with an extrapolation distance parameter $E_{max}$. We sample $m$ 3D points in $\mathcal{B}_+$ separated by a constant distance $\lambda_v$ that is obtained by dividing the smallest edge of $\mathcal{B}_+$ by a fixed resolution parameter $r_v$. Using this method we ensure that $m$ will be of the same order of magnitude for all scenes, avoiding the generation of intractable density volume.

To consider that a given location is unreachable in the scene, we set a threshold $t_\sigma$ and only take in account 3D points with density higher than $t_\sigma$.

## 3.3 Virtual camera locations

Next, our virtual camera location generation algorithm takes poses of real training images as input, but also the NeRF density volume and the desired number of virtual cameras $n$. Our method is two-step: defining the virtual camera positions and then determining their orientations. Our focus is to generate a training dataset for a pose regressor, therefore we want a set of locations to be uniformly distributed all over the area and viewpoints that could be visited by the agent to localize.

To generate our virtual camera positions candidates, we use a similar strategy as the one described in the previous section: we sample $n_i$ 3D points in $\mathcal{B}_+$ using a constant distance $\lambda_s$ between the 3D points obtain by dividing the smallest edge of $\mathcal{B}_+$ by a resolution parameter $r_i$, $i \in \mathbb{N}$. Then we proceed to a multiple criteria pruning step to remove irrelevant candidates: 3D points closer to $d_\sigma$ to a 3D point from the density volume are considered too close to a structure and are discarded, 3D points further than $d_{max}$ to a real camera view position are considered too far from the area of interest and are also discarded. We implement the nearest neighbour search with KDTrees for efficiency.

After candidate pruning, we obtain a final number of virtual camera positions $\hat{n}_i$. If $\hat{n}_i$ is smaller than desired number of cameras $n$, we update the resolution parameter $r_i$ and repeat the generation procedure: $r_{i+1} = r_i + \sigma_r$, with $\sigma_r$ the update step.

Finally, we need to define a camera orientation $(q_x, q_y, q_z, q_w)$ attached to the camera center $(x, y, z)$ for each virtual camera. In order to avoid degenerated views, we copy the orientation of the nearest pose in training set and add a small random perturbation on each axis drawn uniformly in $[-\frac{\theta}{2}, \frac{\theta}{2}]$, where $\theta$ is the maximum amplitude of the perturbation. As an output, we have a set of $n$ poses $[x, y, z, q_x, q_y, q_z, q_w]$ that are used as queries for novel view synthesis, see figure 2-c. More examples of generated poses can be find in figure 3.

## 3.4 Novel view synthesis

Novel views are synthesised on each virtual camera location using $\text{MLP}_{\theta_1}$ and $\text{MLP}_{\theta_2}$ (figure 2-d). The appearance embedding $\mathcal{L}_r^{(a)}$ is chosen by random interpolation of training set appearances.

As a result, rendered images depict a scene from chosen viewpoints where transient occluders observed in training images are removed. Appearance interpolation acts as a data augmentation technique that increases robustness of the localization model under varying illuminations. Some example of resulting appearances are shown in figure 3 and detailed in supplementary materials.

## 3.5 Camera pose regressor training

The final step of our pipeline is to train the localization algorithm with our synthetic-augmented dataset. We simply combine real images with our synthesised images together; Then mini-batches are sampled randomly in this combined dataset for training stage (figure 2-d). We use CoordiNet [8] as a camera pose regressor. CoordiNet is a CNN that uses inductive biases in the network architecture that facilitate geometrical reasoning for the localization task, and also provides an uncertainty quantification for each pose that can be used as a covariance matrix.

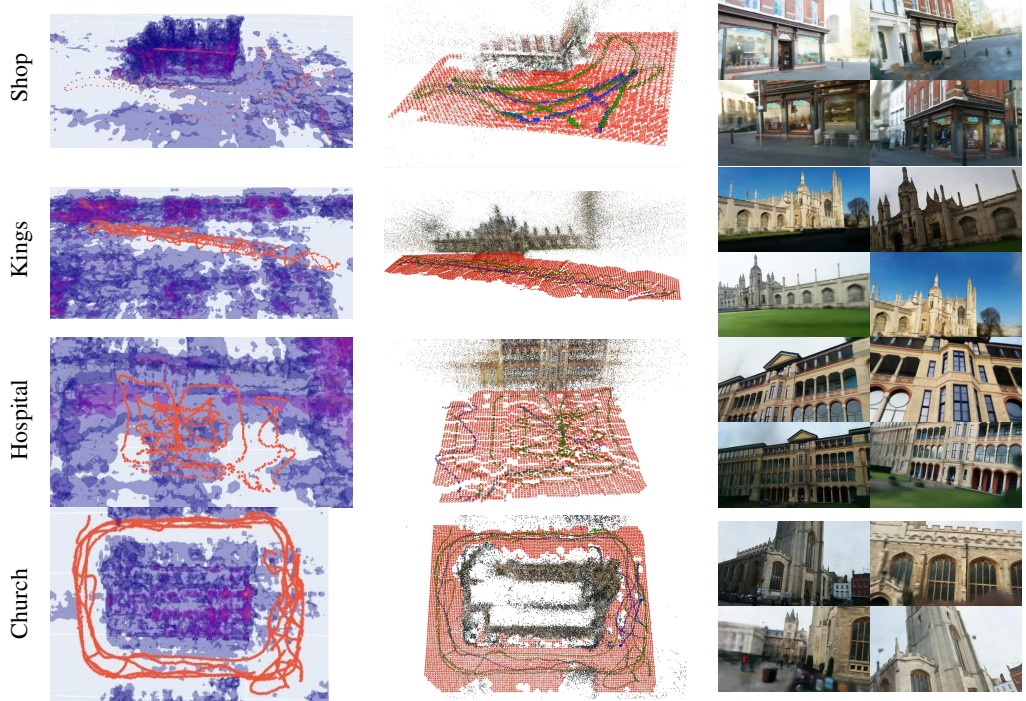

Figure 3: Visualisation of density volumes, virtual camera queries (training poses, test poses, virtual cameras) and example of rendered images on Cambridge Landmarks.

## 4 Experiments

In this section we evaluate LENS combined with CoordiNet on standard localization benchmarks, conduct an ablation study to confirm our design choices and investigate on training poses distribution for pose regressor training.

### 4.1 Datasets and implementation details

**Datasets.**    We evaluate LENS on two standard visual localization benchmarks:

- **Cambridge Landmarks** [7] contains 4 outdoor dynamic scenes captured by a smartphone. In each scene, a common building is visible from each image. Camera poses are recovered from SfM and training sets contains between 200 and 2000 images extracted from videos. We downscale input images to 640×360 pixels.
- **7scenes** [28] consists of 7 static indoor scenes, captured by a Kinect RGB-D sensor. This dataset is challenging for camera pose regression algorithms because test sequences follow different paths than the ones of train sequences. We downscale input images to 320×240 pixels.

**NeRF parameters.**    We use the code of [29] to train one NeRF-W [27] by scene, using the default parameters of the Pytorch implementation. We train each model for 20 epochs using the training images to generate the training rays. We took a maximum of 5 training sequences (5k images) for the 7 scenes dataset and we define one appearance embedding by image in the training set. Generating images of Cambridge scenes takes approximately 40s/GPU, with 256 coarse and 256 fine sampling by ray. It takes 7s/GPU for 7 scenes while sampling only 128 coarse and 128 fine values as the scenes are smaller and less complex than Cambridge scenes.

**Virtual camera locations generation.**    For outdoor scenes, virtual camera locations are sampled on a 2D plane and LENS parameters are: $r_v = 128$, $t_\sigma = 20$, $d_{max} = 8m$, $d_\sigma = 1m$, $E_{max} = 1m$,

$r_0 = 1$, $\sigma_r = 1$ and $\theta = 15°$. For indoor scenes, the regular grid is defined on 3 dimensions and we adapt the following LENS parameters: $d_{max} = 50cm$, $d_\sigma = 20cm$, $E_{max} = 20cm$ and $\theta = 20°$.

We set the desired number of virtual views $n$ to 500% for outdoor scene and 1000% for indoor scene of the total number of real training images. We found that amount of images a good trade-off between computational cost and localization accuracy. More details are provided in section 4.3.

**Pose regressor.** We use CoordiNet [8] as camera pose regressor with EfficientNet-b3 [30] as backbone and optimize the network by maximizing heteroscedatic log-likelihood (see [8]). We train our models for 250 epochs with a fixed learning rate of $1e - 4$ for both datasets. During training, we use a batch size of 10 for Cambridge Landmarks and 40 for 7 scenes.

## 4.2 Comparison with related localization methods

**Competitors.** We compare our method CoordiNet + LENS with CoordiNet only trained on real images. SPP-Net [22] is a small CNN pose regressor that has been trained with and without additional synthetic data, providing a good baseline for LENS. We also report results from TransPoseNet[11], which is a state of the art transformer approach for camera pose regression. DirectPoseNet [16] uses a NeRF model as a photometric supervisor during the training. Active search [31] is a baseline for structure-based methods, where local image features are matched against a 3D point cloud obtained by SfM.

| Dataset | Camera Pose Regression | | | CPR + view synthesis | | | 3D |
|---|---|---|---|---|---|---|---|
| Cambridge | SPPNet [22] | CoordiNet [8] | TransPN [11] | DirectPN [16] | SPPNet + synth [22] | **CoordiNet + LENS** | Active search [31] |
| K College | 1.91/2.4 | 0.70/0.9 | 0.60/2.4 | - | $0.74^{61}/1.0^{58}$ | $\mathbf{0.33^{53}/0.5^{44}}$ | 0.42/0.6 |
| OldHosp | 2.51/3.7 | 0.97/2.1 | 1.45/3.1 | - | $2.18^{13}/3.9^{5}$ | $\mathbf{0.44^{55}/0.9^{57}}$ | 0.44/1.0 |
| Shop | 1.31/7.8 | 0.69/3.7 | 0.55/3.5 | - | $0.59^{55}/2.5^{68}$ | $\mathbf{0.27^{61}/1.6^{57}}$ | 0.12/0.4 |
| Church | 3.21/7.0 | 1.32/3.6 | 1.09/5.0 | - | $1.44^{55}/3.3^{53}$ | $\mathbf{0.53^{60}/1.6^{56}}$ | 0.19/0.5 |
| Average | 2.24/5.2 | 0.92/2.6 | 0.91/3.5 | - | $1.24^{45}/2.7^{48}$ | $\mathbf{0.39^{58}/1.2^{54}}$ | 0.29/0.6 |
| 7scenes | | | | | | | |
| Chess | 0.22/7.6 | 0.14/6.7 | 0.08/5.7 | 0.10/3.5 | $0.12^{45}/4.4^{42}$ | $\mathbf{0.03^{79}/1.3^{80}}$ | 0.04/2.0 |
| Fire | 0.37/14.1 | 0.27/11.6 | 0.24/10.6 | 0.27/11.7 | $0.22^{41}/8.9^{37}$ | $\mathbf{0.10^{63}/3.7^{68}}$ | 0.03/1.5 |
| Heads | 0.22/14.1 | 0.13/13.6 | 0.13/12.7 | 0.17/13.1 | $0.11^{50}/8.3^{41}$ | $\mathbf{0.07^{63}/5.8^{57}}$ | 0.02/1.5 |
| Office | 0.32/10.0 | 0.21/8.6 | 0.17/6.3 | 0.16/6.0 | $0.16^{50}/5.0^{50}$ | $\mathbf{0.07^{67}/1.9^{78}}$ | 0.09/3.6 |
| Pumpkin | 0.47/10.2 | 0.25/7.2 | 0.17/5.6 | 0.19/3.9 | $0.21^{55}/4.9^{52}$ | $\mathbf{0.08^{68}/2.2^{69}}$ | 0.08/3.1 |
| Kitchen | 0.34/11.3 | 0.26/7.5 | 0.19/6.8 | 0.22/5.1 | $0.21^{38}/4.8^{58}$ | $\mathbf{0.09^{65}/2.2^{71}}$ | 0.07/3.4 |
| Stairs | 0.40/13.2 | 0.28/12.9 | 0.30/7.0 | 0.32/10.6 | $0.22^{45}/7.2^{45}$ | $\mathbf{0.14^{50}/3.6^{72}}$ | 0.03/2.2 |
| Average | 0.33/11.6 | 0.22/9.7 | 0.18/7.8 | 0.20/7.3 | $0.18^{45}/6.2^{47}$ | $\mathbf{0.08^{64}/3.0^{69}}$ | 0.05/2.5 |

Table 1: **6DOF localization errors of visual localization methods.** We report median translation/orientation error (meters/degrees). TransPN and DirectPN stand for TransPoseNet and DirectPoseNet. Superscripts numbers refer to the relative improvement (green) or deterioration (red) in percentage brought by synthetic data.

**Results.** Median translation and orientation errors are reported in table 1. CoordiNet + LENS achieves best reported results for camera pose regression methods on both Cambridge Landmarks and 7scenes with a large margin. Moreover, our method is more accurate than Active Search [31] on 5 scenes out of 11, reducing the gap between camera pose regression and structure-based methods. Then, we observe that LENS provides better relative localization improvement (approximately +60%) than the synthetic datasets generated by Purkait et al. [22] (+45%). Visualizations and comparison of resulting trajectories are provided in supplementary materials.

## 4.3 Ablation study

**Synthetic dataset size.** We change the amount of generated sample from 100% to 1000% (up to 5000% for fire scene) of the total number of images in the real training set on three different scenes and we compare the relative improvement in term of localization accuracy. Results are shown in table 2.

| Synth. im. / Real im. | **ShopFacade** (231 im.) | **Church** (1487 im.) | **Fire** (2000 im.) |
|---|---|---|---|
| **0%** | 0.61/4.2 | 1.06/3.1 | 0.27/11.6 |
| **100%** | 0.61/2.4 | 0.75/2.4 | 0.18/5.7 |
| **200%** | 0.41/2.0 | 0.58/1.9 | 0.16/5.7 |
| **500%** | 0.26/1.2 | 0.51/1.5 | 0.12/4.2 |
| **1000%** | 0.25/1.2 | 0.45/1.6 | 0.10/3.2 |
| **5000%** | - | - | 0.07/2.4 |

Table 2: median translation (m) and orientation (°) errors depending on synthetic dataset size

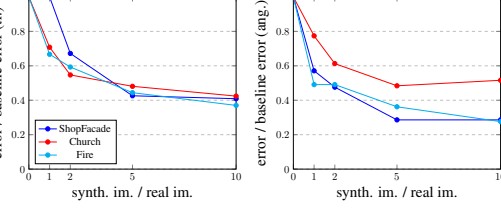

Figure 4: Translation (tr., left) and angular (ang., right) error relative decrease vs synthetic dataset size.

As expected, we observe that using a higher grid resolution containing more samples leads to a better localization. In addition to that, we can see in figure 4 that the relative improvement compared to the baseline (*i.e.* no synthetic images) is correlated to the ratio between synthetic and real images rather than the total number of images itself: a ratio of 10 leads to a 59% translation improvement for ShopFacade, 58% for Church and 63% for Fire. Curves of error translation in figure 4 do not seems to reach a plateau: an higher ratio would potentially bring better localization results.

**Benefit of using volume and random appearances.** In table 3, we observe that occluded views located too close or inside a building (shown in supplementary materials) disturb the training of the pose regressor, decreasing the localization accuracy. We can see in figure 5 that the use of the density volume provided by the trained NeRF correctly remove distracting samples. Providing a random appearances embedding during the NeRF image generation also decrease slightly translation and rotation errors. Even if the improvement is minor and not very significant on this scene, we expect this appearance augmentation to be very useful on experiments with more diversity in illuminations (day and night images for example).

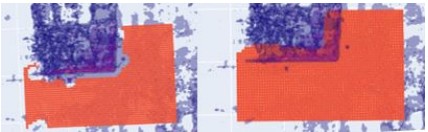

| Errors | Volume | | Appearances emb. | |
|---|---|---|---|---|
| | with | without | random | constant |
| Translation | 0.27m | 0.36m | 0.25m | 0.26m |
| Rotation | 1.6° | 1.7° | 1.2° | 1.4° |

Figure 5: virtual camera locations with (left) and without (right) NeRF volume pruning step.

Table 3: Localization errors comparison for density volume and appearances embedding ablations on Shop Facade scene.

## 4.4 Exploring camera pose regression on synthetic domain

In order to investigate the impact of training camera poses distribution on the performance of pose regression without taking in account the domain gap between real and synthetic data, we perform the following experiment: we replace both training and testing real images by NeRF-rendered images at the exact same location, then we trained and test a pose regressor on this synthetic dataset. Results are reported for the indoor scene fire in the two first columns of table 4 (seq3-4 real vs seq3-4 synth.). We get similar localization performances compared to the same model trained on real data, which allow us to perform deeper analysis on the impact of distribution of camera poses used for the training.

| **Method** | seq3-4 | seq3-4 | LENS | LENS | LENS | Act. Search [31] |
|---|---|---|---|---|---|---|
| **Data type** | real | synth. | synth. | synth. | synth. | real |
| **Train set size** | 2k | 2k | 2k | 10k | 100k | - |
| **Error (m/°)** | 0.27/11.7 | 0.27/10.6 | 0.08/3.5 | 0.05/2.4 | 0.03/1.4 | 0.03/1.5 |

Table 4: Median translation (m) and orientation (°) errors in Fire scene from 7scenes. seq3-4 refers to methods using images from sequences 3 & 4 as training data.

In a second experiment, we replace the original training camera poses by uniformly distributed poses generated by LENS. We observe a important decrease in median localization error from 27cm/11.7° to 8cm/3.5° whereas we use the same number of training data (table 4 columns 2 vs 3). Increasing the number of synthetic data, up to 5000% of the number of training data in the original training set,

leads to even better localization results (table 4 columns 3 to 5). We also show that pose regression can reach structure-based method accuracy (evaluated on real data) if the model is trained with the largest amount of data. A similar experiment had been made by Sattler et al. [12] with opposite conclusions: even with a big synthetic dataset depicting the scene, pose regressors were not able to reach an accuracy comparable to structure-based methods, suggesting that camera pose regression algorithms are inherently limited by their approach without geometrical constraints.

From these experiments, we end up with a different conclusion: datasets using only video sequences create an unbalanced regression problem where training labels does not cover the entire set of possible poses and then lead to a poor localization accuracy. Deep learning approaches are known to perform well in high data regimes, but the main finding of this research is that training datasets distributed across the entire scene are crucial for camera pose regression performances. The different results we observe compared to [12] probably come from a higher quality of our synthesized views and a pose regression architecture that performs better.

### 4.5 Limitations

The first limitation of our pipeline is the offline computation time: Nerf-W needs to be trained several days with a GPU on a single scene in order to reach optimal rendering results. The slow rendering of novel views forces us to generate a dataset before training the localization algorithm instead of an online data generation pipeline. Faster synthesis would enable to learn from an infinite quantity of data instead of a limited number of images. However, faster NeRF training [32] and rendering [33, 34] are active research fields that should enhance speed and performances of LENS in the future. Moreover, as reported in Table 5, our method offsets the costly offline computation by a fast and light online localization, enabling real-time embedded applications for robotics.

We also observed that training with only synthetic images leads to poor performances on real images. This is due to the domain gap between real and synthetic images: no dynamic objects, different textures, more blur and some artifacts are observable in the synthetic domain. Mixing real and synthetic images is the simpler way to mitigate this issue, however domain adaptation techniques [35] could be used to reduce domain discrepancy as well as higher quality rendered samples.

| | | Offline (1 scene) | | | | Online (1 image) | |
|---|---|---|---|---|---|---|---|
| **Structure based** | Task | SfM | | Quantization | | SIFT | Loc. |
| | Timing | Hours* | | Minutes* | | 50-500ms | ∼500ms* |
| | Memory | NR | | NR | | NR | GBs*† |
| **LENS + CoordiNet** | Task | SfM | Train NeRF | NVS | Train PR | Coordinet inference | |
| | Timing | Hours* | Hours* | 30s/im/gpu* | Hours* | ∼50ms | |
| | Memory | NR | NR | NR | NR | <50MB | |

Table 5: Approximate time and memory requirements comparison between structure-based methods and ours. * denotes a scaling time and memory consumption according to the scene size. † structure-based methods need to load the features vocabulary and access to the 3D points cloud during localization.

## 5 Conclusion

In this paper, we address limitations of camera pose regressors at data-level: thanks to high quality novel view synthesis provided by NeRF-W, we propose to train a relocalization algorithm with synthetic images uniformly sampled on the entire scene. Our method enhance strongly localization performances, reducing the gap with structure based methods while keeping the advantages of pose regression: a fast inference with low memory footprint and the capability to scale to large environments. Our experiments show that camera pose regression can perform well when trained with large and diverse datasets. LENS coupled with CoordiNet improves camera pose regression state of the art and can be used for accurate real-time robot relocalization system. As novel view synthesis is an active research field, we expect to improve LENS with faster rendering and higher quality samples as a future direction.

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
