# OpenReview forum: "LENS: Localization enhanced by NeRF synthesis"
_robot-learning.org/CoRL/2021/Conference — CoRL2021 Poster_

### Official Review · Reviewer_BBFW · 2021-07-02

**Originality:** Good
**Technical Quality:** Very Good
**Clarity Of Presentation:** Excellent
**Impact:** 4

**Recommendation:**

Strong Accept: I recommend accepting the paper and will argue for my recommendation even if other reviewers hold a different opinion.

**Summary:**

The paper presents an approach named LENS as an offline data augmentation technique for camera pose relocalization problems. A common mode of failure for camera pose relocalization approaches is the varying distribution between the training set and the test set which prevents the model from being able to reasonably extrapolate the pose information for a newly observed image. This work addresses this problem by proposing to leverage novel view synthesis as an offline intermediate step to generate synthetic views of the scene. A pose regressor method is then trained on a combination of the real and synthetic dataset demonstrating a substantial performance improvement. Several ablation experiments were conducted offering insights towards the underlying problem and the proposed solution.

**Issues:**

See weaknesses / questions above.

**Reviewer Expertise:**

Excellent: Expert knowledge on the topic of the paper

**Strengths And Weaknesses:**

Strengths:
- The paper is well written, it follows a clear line of thought from problem to proposed solution. It is indeed enjoyable to read.
- The proposed solution addresses an important problem and a weakness for a large number camera pose relocalization methods.
- The results demonstrate that the use of synthetic images results achieving the same performance as  3D-based methods.

Weaknesses / Questions:
- The idea of generating virtual views for training is not entirely novel. See [1]. While the application area differs, the work should also be cited.
- How are poses close to occluders discarded?
- For the virtual camera location generation, how were the values of d_max and d_sigma selected? And how does one know if the pose is "too close to a structure" or "too far away"?
- Current sampled virtual views are quite dense. Are they all required for the pose regressor to perform well? Instead of rendering on an evenly spaced grid can one generate virtual views in a more concise pattern to achieve the same results?
What is your hypothesis behind the contradictory results in table 4 between your results and the results from Sattler et al? Is it due to the better distribution of the training images that the model can better capture the structure? Are the training images different here and in Sattler et al?
- What are the failure cases for the localization? In the video at 01:26 there's a much larger gap between the model predictions and the ground-truth. What was the cause of that?


[1] Abhijit Kundu, Xiaoqi Yin, Alireza Fathi, David Ross, Brian Brewington, Thomas Funkhouser, Caroline Pantofaru. Virtual Multi-view Fusion for 3D Semantic Segmentation. In ECCV 2020.

**Summary Of Recommendation:**

While the proposed solution of generating synthetic views of the scene and using them for training may not be novel in its entirety, this is the first approach demonstrating that using views generated from NVS methods can be beneficial for the trained model while backing it up with ablation experiments.

---

> ### Author Response · Authors · 2021-08-24
> **Authors reply to reviewer BBFW**
>
> We sincerely thank the reviewer for the useful comments and interesting questions.
>
> •	The relevant citation mentioned by the reviewer about a similar work for 3D semantic segmentation has been added to the related work.
>
> > How are poses close to occluders discarded ?
>
> To do so, we use the density volume learned by the NeRF model to avoid static objects like building and urban furniture. Details are provided in section 3.2 and 3.3
>
> >  For the virtual camera location generation, how were the values of d_max and d_sigma selected? And how does one know if the pose is "too close to a structure" or "too far away"?
>
> These hyperparameters have been selected based on scenes type (8m and 1m for outdoor, 0.5m and 0.2m for indoor). We tried different values, but observed few performances improvement or decrease, suggesting that these parameters are not crucial for our method, as long as we don’t use extreme and non-meaningful values.
>
> > Current sampled virtual views are quite dense. Are they all required for the pose regressor to perform well? Instead of rendering on an evenly spaced grid can one generate virtual views in a more concise pattern to achieve the same results?
>
> We agree that we use dense sampled views, requiring a long offline rendering process. Figure 4 shows that this high density is important to obtain optimal results but also that even with less density, the method brings localization improvement thanks to the uniform distribution across the scene. This “ad-hoc” method to sample virtual camera locations is probably not optimal from an efficiency point of view and could be improved in a future work, however it helps to support our claim: training sets distributed uniformly on the scene are crucial for camera pose regression.
>
> > What is your hypothesis behind the contradictory results in table 4 between your results and the results from Sattler et al? Is it due to the better distribution of the training images that the model can better capture the structure? Are the training images different here and in Sattler et al?
>
> Our first hypothesis for these contradictory results is that we use a different novel view synthesis method that renders higher quality samples, enabling better localization performances. Moreover, we also use a more recent pose regression architecture. The virtual camera locations were selected with approximately the same method, but we are able to generate synthetic data further away of the poses of real training images than them (8m vs 3m) as we can discard disruptive samples using the NeRF volume.
>
> > What are the failure cases for the localization? In the video at 01:26 there's a much larger gap between the model predictions and the ground-truth. What was the cause of that?
>
> This interesting failure case happens because the camera holder changes the viewpoint to observe the top part of the church. We suppose that such a viewpoint from this location has not been seen during training (in both real and synthetized views), but a similar viewpoint was captured from another location few meters away, resulting in a failure case. This could be solved by adding more diversity in synthetic camera orientations.

---

### Official Review · Reviewer_HedN · 2021-07-12

**Originality:** Very Good
**Technical Quality:** Very Good
**Clarity Of Presentation:** Good
**Impact:** 4

**Recommendation:**

Weak Accept: I recommend accepting the paper, but will not argue for my recommendation if the majority of other reviewers have a different opinion.

**Summary:**

This paper proposes to use images synthesized by NeRF as augmented data to improve camera pose regression. Specifically, authors find that deciding where to render images based on NeRF's density volume can further reduce the error since it can avoid views that are too close/far from the scene. Experiments show that training the same pose regression method with synthetic dataset generated by NeRF can dramatically reduce the error.


**Issues:**

- Can you attach the memory footprint and runtime of both CoordiNet[8] and Active search [30]? Currently, the text only says they are slow but it would be nice to show the actual comparison.

-  I think Figure 3 can be further improved by 1) making the camera queries' colors more obvious and 2) adding (left, middle, right) in the caption to emphasize the meaning of different columns.

- I don't think equation 1 is really needed. The text description "... random interpolation of training set" is already clear.

- Line 246, “is train with” -> “is trained with”.

- Line 267, “artefacts” -> “artifacts”.

- Line 113, “see figure 2-a” maybe can be moved to line 112 because it shows how the models are wired instead of the registered images.

- Line 128 “the poses,” seems to be redundant.


**Reviewer Expertise:**

Very good: Comprehensive knowledge of the area

**Strengths And Weaknesses:**

Strengths:
The idea is simple and sound, I like it!
The performance boost is huge.
The ablation study on the synthetic dataset size is a useful reference for the literature. The results clearly show that regression-based methods can work well when the data is abundant.

Weaknesses:
The procedures of training NeRF and generating images are time-consuming. However, as mentioned by the authors, they may be addressed as the NeRF literature develops.


**Summary Of Recommendation:**

I recommend accepting the paper because the idea of using NeRF for data augmentation is sound, the proposed strategy for placing virtual cameras is shown to be helpful, and the performance improvement is large. The paper can be more clearly written based on the issues I point out below but overall it is already complete.

---

> ### Author Response · Authors · 2021-08-24
> **Authors reply to reviewer HedN**
>
> We sincerely thank the reviewer for the useful comments. Below we address the issues raised in the review:
>
> •	A new table has been added in the paper to compare runtime and memory footprint of CoordiNet and structure based methods (e.g. Active Search). This table exhibits our runtime advantage during the online localization process at the cost of more offline processing.
>
> •	Figure 3 has been modified as suggested.
>
> •	typos and redundancies pointed out have been corrected

---

### Official Review · Reviewer_CXrf · 2021-07-23

**Originality:** Good
**Technical Quality:** Very Good
**Clarity Of Presentation:** Very Good
**Impact:** 3

**Recommendation:**

Weak Accept: I recommend accepting the paper, but will not argue for my recommendation if the majority of other reviewers have a different opinion.

**Summary:**

This paper proposes a method for pose regression that makes use of neural
radiance fields (NeRFs) as an intermediate representation. In the proposed
method, first a NeRF model of a scene is constructed from a set of images with
known reference poses (e.g. as provided by a structure from motion pipeline).
The NeRF model can be queried at a variety of poses to produce synthetic images,
which are subsequently used (in conjunction with the original training data) to
train a pose regression network. Finally, at test time, the pose regressor
network can be queried online with new images to infer poses. The key insight
being leveraged by the proposed approach is that NeRF models can be used to
produce (in principle) an arbitrarily large amount of good enough synthetic
images to offer clear improvements to the accuracy of the pose regression
module. Pruning heuristics are provided to ensure the selected query viewpoints
have "good" geometric properties for pose regression. Experimental results
suggest not just that the approach improves over existing pose regression work,
but also that the pruning heuristics are important for mitigating degenerate
images obtained from the NeRF model.


**Issues:**


## Clarifying Contribution

- The contribution should be clarified a bit (see Strengths and Weaknesses) to
  clarify the distinction between this approach and similar work and potential
  advantages. It seems like the fact that this method can be faster (I am
  inferring this based on the run time of CoordiNet) than strictly NeRF-based
  pose inference is undersold relative to the result that the approach is more
  accurate than pose regression without the NeRF model. Both seem (to me) to be
  important.

## Experimental Results

- It would be helpful to additionally report test-time computation speed for the
method (not necessarily for each dataset, etc, but even just a mention in the
text to give an idea); it seems this should be similar to CoordiNet, but good to
report this in any case, as it is a possible advantage relative to other
NeRF-based localization methods.

## Questions and Comments

- Most of the examples considered contain scenes where the camera motion is
  constrained to be more-or-less planar. I could see the data generation process
  for this approach becoming impractical for large scale scenes with 3D camera
  motions (e.g. for images from aerial vehicles); I'd be curious about the
  authors' thoughts on the scalability of the method to these types of
  environments (Section 4.5 mentions faster NeRF training and rendering, but the
  number of query images for this approach still seems to scale with the
  dimension of the environment - let me know if I'm misunderstanding something).

- In Figure 1, it might help to mark which of the images shown are training
images and which are synthetic (it wasn't entirely clear to me from the arrows).

- Some more detail on the training process (e.g. cost function used, etc.) would
be helpful.

- Rotations in Table 3 use a decimal comma, whereas in Table 2,4 they use a
  decimal period. Table 3 uses centimeters, other tables use meters (this is OK,
  but I don't see a reason not to just keep the units the same?)

- Some references are not formatted correctly (e.g. "inerf" -> "iNeRF") or
missing information e.g. [8], [11], [15], and others.


**Reviewer Expertise:**

Good: General knowledge of the area

**Strengths And Weaknesses:**


## Strengths

This is a well-written and organized paper, and implicit neural map
representations are definitely relevant.

Conceptually, I found this to be a really interesting approach. The approach of
training a pose regression module offline using the rendered output of a NeRF
model for subsequent use in an online fashion nicely circumvents the issue of
slow rendering with present NeRF techniques (at the expensive of a
time-consuming offline training process).

I hadn't thought about the issue of pruning degenerate images, but it is
interesting that this offered a measurable (albeit small?) improvement in the
accuracy as in Section 4.3.

As a remark, there is a neat analogy between the overall approach here and the
classical approach to Monte Carlo localization (e.g. in Probabilistic Robotics
by Thrun et al.). In practice, rather than computing an inverse sensor model at
run-time, one will pre-compute a cached representation of the model for lots of
possible observations and look up / interpolate between cached observations
online (as in Chapter 8.2.3 of Probabilistic Robotics). It seems like this
approach is essentially doing something similar: where a method like iNeRF (ref.
[15]) would regress directly from the NeRF model to an SE(3) pose, the proposed
approach densely queries the NeRF model to construct an implicit intermediate
"cached" representation to approximate what would otherwise be the
maximum-likelihood estimate provided by an iNeRF style model, but which is
faster to query than the NeRF model itself.

The experimental results for the paper were very thorough. I particularly
appreciated the ablation study and the supplementary results (+ video!). Very
nice!

## Weaknesses

While the paper does a good job demonstrating improvements over prior work on
pose regression, it's not made explicit in the paper as presented what
advantages the proposed approach confers in comparison to a method like iNeRF
(ref. [15]), where the NeRF model is used to directly infer the camera pose
rather than training an intermediate pose regressor (let me know if I am missing
something, though). One thought is that this approach should be faster
(specifically at test time) than iNeRF, since only the pose regressor is used at
runtime (thereby eliminating explicit runtime dependence on the NeRF model), and
CoordiNet itself is capable of running at ~ 18 Hz. This is a distinction that I
really feel should be made more clear; to me, this would be the most blatant
advantage of this method with respect to those techniques!

More broadly, the approach of querying a NeRF model from lots and lots of poses
seems a bit "brute force." From my reading, the goal is essentially to translate
from one implicit scene representation (the NeRF model) to another (the pose
regression). I wonder if this could be done in a more direct way. For example,
while the related work section suggests that this sort of "ad hoc" method for
view selection is sufficient, maybe an alternative approach could help reduce
the number of views needed for training.

**Summary Of Recommendation:**

I am recommending a "weak accept": I think this is a really interesting
contribution, and has some potential advantages over existing NeRF-based
localization and current pose regression work. To improve the paper, I recommend
clarifying the contribution with respect to strictly NeRF-based localization and
adding a bit to the discussion of scalability (see Issues).

---

> ### Author Response · Authors · 2021-08-24
> **Authors reply to reviewer CXrf**
>
> We sincerely thank the reviewer for the useful comments. Below we address the questions raised:
>
> •	The strengths of our approach compared to other type of visual localization methods are definitely low computation time and memory footprint during online localization compared to structure based methods, which make it more convenient for robotics real-time embedded applications. An additional table has been added in the paper to exhibit this strength. Methods like iNeRF do not match this real-time constraint because of the gradient descent optimization and many queries to the NeRF that have to be performed at test time. The authors report 20 seconds spent only on gradient descent optimization to locate an image.
>
> •	We agree that our method is more adapted on scenes where the camera motion is planar. However, the 7scenes dataset contains indoor scenes where the camera moves in the entire 3D space (see supplementary materials) and our method still performs well. In this case, we sample virtual camera locations on a 3D grid which require more views than in the 2D case.
>
> •	All images that are shown in Fig 1 are synthesized by NeRF-W, the caption has been updated.
>
> •	About the loss function, we mention in 4.1 that we optimize the network by maximizing log-likelihood following the process described in CoordiNet paper. However, we suppose that any loss function commonly used for camera pose regression should work with LENS.
>
> •	Useful remarks about units in tables and references have been taken into account and corrected.

---

### Meta-Review · Area_Chair_egYJ · 2021-08-12

**Recommendation:** Accept (Poster)
**Confidence:** 3

**Metareview:**

The reviews are generally positive. The AC endorses the reviewers' recommendation. The authors are encouraged to address questions and concerns raised in the reviews.

---

### Decision · Program_Chairs · 2021-09-13

**Decision:**

Accept (Poster)

**Comment:**

The reviews are generally positive. The AC endorses the reviewers' recommendation. The authors are encouraged to address questions and concerns raised in the reviews.